# Silybin Modulates Collagen Turnover in an In Vitro Model of NASH

**DOI:** 10.3390/molecules24071280

**Published:** 2019-04-02

**Authors:** Beatrice Anfuso, Pablo J. Giraudi, Claudio Tiribelli, Natalia Rosso

**Affiliations:** Fondazione Italiana Fegato, ONLUS, AREA Science Park Basovizza SS 14 km 163.5, 34149 Trieste, Italy; b.anfuso@fegato.it (B.A.); pablo.giraudi@fegato.it (P.J.G.); ctliver@fegato.it (C.T.)

**Keywords:** NAFLD, NASH, Silybin, fibrogenesis, coculture model, hepatic stellate cells

## Abstract

Silybin has been proposed as a treatment for nonalcoholic steatohepatitis (NASH). In this study, we assessed the effect of Silybin in a well-established in vitro coculture model of early-stage NASH. LX2 and Huh7 cells were exposed to free fatty acid (FFA) and Silybin as mono- or coculture (SCC). Cell viability, LX2 activation, collagen deposition, metalloproteinase 2 and 9 (MMP2-9) activity, and ROS generation were determined at 24, 96, and 144 h. Exposure to FFA induced the activation of LX2 as shown by the increase in cell viability and upregulation of collagen biosynthesis. Interestingly, while cotreatment with Silybin did not affect collagen production in LX2, a significant reduction was observed in SCC. MMP2-9 activity was reduced in FFA-treated Huh7 and SCC and cotreatment with Silybin induced a dose-dependent increase, while no effect was observed in LX2. Silybin also showed antioxidant properties by reducing the FFA-induced production of ROS in all the cell systems. Based on these data, Silybin exerts its beneficial effects by reducing LX2 proliferation and ROS generation. Moreover, MMP2-9 modulation in hepatocytes represents the driving mechanism for the net reduction of collagen in this NASH in vitro model, highlighting the importance of hepatic cells interplay in NASH development and resolution.

## 1. Introduction

The increasing trend of obesity worldwide is accompanied by the increasing prevalence of many obesity-related disorders such as cardiovascular and kidney diseases, diabetes, cancers, and musculoskeletal disorders [1]. Nonalcoholic fatty liver disease (NAFLD) is tightly associated with obesity and it is becoming the leading cause of liver-related mortality [2]. NAFLD encompasses a wide spectrum of conditions, from simple accumulation of fat (steatosis) to steatohepatitis (NASH), fibrosis, and cirrhosis with consequent higher risk to develop hepatocellular carcinoma (HCC) [2,3,4]. The accumulation of extracellular matrix in the liver, which leads to progressive fibrosis, cirrhosis and liver failure, is the major cause of liver-related death in patients with NAFLD [5,6]. An early event in the development of hepatic fibrosis is the activation of hepatic stellate cells (HSC). Upon activation, HSC change their phenotype into myofibroblast-like cells with increased production of extracellular matrix (ECM) components, with the production of cytokines and chemokines for the recruitment of inflammatory cells, and the progression from fatty liver to NASH [7,8]. To date, despite many efforts to unravel NAFLD pathogenesis, no effective agent is available for the treatment of fibrosis in NAFLD context. 

The extract from milk thistle fruit and seeds is referred to Silymarin, which is a mixture of polyphenolic flavonoids. Silybin (also referred to Silibinin) is the major component (50–60%) of Silymarin, and it is the primary bioactive principle. Several studies have shown its hepatoprotective and antihepatotoxic properties in many liver diseases such as viral hepatitis, alcoholic liver disease, hepatic intoxication, cirrhosis, and HCC [9,10]. The antifibrotic efficacy of Silybin is mainly supported by results obtained in animal models where Silybin significantly reduced α-smooth muscle (α-SMA) positive cells and procollagen mRNA and regulated the expression of matrix metalloproteinases [11,12,13]. More recently, studies on in vivo model of NAFLD demonstrated its efficacy also in this disease where Silybin ameliorated insulin resistance, dyslipidemia, inflammation, oxidative stress, and fibrosis with a general improvement of the liver health [14,15,16]. 

In this context, some clinical trials evaluated the efficacy of Silybin. Most of the studies found that Silybin is able to reduce transaminases level as well as induce a general improvement of the “hepatic status” [17,18]. Moreover, a trial performed in adults subjects with biopsy-proven NASH showed that silymarin reduced liver fibrosis and stiffness with no effects on NAFLD score [19]. In other studies, authors found that the combination of Silybin with vitamin E improved hepatic biochemical profile, anthropometric parameters, lipidic and glycemic metabolisms, and the ultrasonographic score of liver steatosis [20,21]. Recently, a meta-analysis involving 587 patients concluded that silymarin represented a promising treatment to improve liver function in NAFLD patients [22]. However, the lack of standardization of Silybin preparations and the combination with other factors (e.g., vitamin E or Mediterranean diet) makes it difficult to take consistent conclusions from these promising data. 

Despite many pieces of evidence about the general hepatoprotective effects of Silybin, both in preclinical models and in a clinical setting, very few data about the molecular mechanisms underlying the antifibrotic efficacy are currently available.

We previously developed an in vitro human NASH model (SCC) able to reproduce the initial phases of NASH development thanks to cell-to-cell interactions. Specifically, we reported that exposure to FFA induced their accumulation into hepatocytes and the activation of hepatic stellate cells as demonstrated by overexpression of α-SMA, accumulation of extracellular collagen and modulation of metalloproteinases (MMPs), and tissue inhibitor of metalloproteinases (TIMP) [23]. Thus, the aim of this study was to assess the in vitro effect of Silybin in our well-established coculture model of NASH that reproduces the initial phases of NASH development and by comparison of the effects on each monoculture system.

Here we demonstrate that Silybin exerts a direct beneficial effect on HSC by reducing the FFA induction of cell proliferation and ROS generation. More importantly, the net reduction of collagen observed in SCC after FFA and Silybin administration was the result of the modulation of MMP2-9 activity on hepatocytes, thus highlighting the importance of the cross-talk between hepatocytes and HSC not only in the development, but also in the resolution of NASH.

## 2. Results

### 2.1. Effects of Silybin on Viability and Cell Proliferation

To assess the safety of our approach, we determined the cytotoxic effect of FFA, Silybin, or the combination of both in the monocultures and SCC by the MTT assay. Silybin alone had no toxic effect at the lowest concentrations at any experimental time; however, at the highest concentration and after 144 h it induced a significant reduction in cell viability in the monocultures (−22%, *p* < 0.01 for Huh7 and −18 %, *p* < 0.05 for LX2) (Figure 1A,B). 

Interestingly, when these two cell types were together (SCC) the toxic effect disappeared (Figure 1C). Exposure to FFA did not alter cell viability either in Huh7 or in SCC, while induced a progressive increase in cell viability along the time in LX2 (144 h, *p* < 0.01 vs. CTRL), indicative of an increase in cell proliferation. Combination of FFA + Silybin did not affect the cell viability in SCC. However, in Huh7 monoculture, it was observed a deleterious effect after exposure to 7.5 µM Silybin (*p* < 0.01 and *p* < 0.05 at 96 and 144 h, respectively, vs. CTRL and FFA) (Figure 1A). Whereas in LX2 the FFA-induced effect was progressively attenuated (144 h, *p* < 0.05 vs. FFA) by the presence of Silybin.

### 2.2. Effect of Silybin on Hepatic Stellate Cells Activation and Collagen Biosynthesis

Hepatic stellate cell (HSC) activation is characterized by morphological and functional changes. Upon activation, HSCs acquire a fibroblast phenotype with changes in the expression of different genes, such as ACTA2 (α-SMA), which is considered one of the activation markers. Activated HSCs show an increase in type I collagen production, which has been associated with fibrogenesis [7]. In this study, we evaluated, both in SCC and LX2 monoculture, the effect of Silybin exposure in counteracting the FFA effect on HSCs activation. 

Regarding the expression of ACTA2 (α-SMA), we did not observe significant changes in the gene expression either in SCC (Figure 2A) or in LX2 (Figure 2B) at any of the experimental concentrations or time.

Despite being not statistically relevant, LX2 cells treated with FFA showed an increasing trend in ACTA2 (α-SMA) expression until 96 h and a sudden drop after 144 h. Concerning collagen biosynthesis, FFA treatment was able to induce the upregulation of COL1A1 in both SCC and LX2 starting from 96 h of treatment (*p* < 0.05 and *p* < 0.001 for SCC and LX2, respectively). While cotreatment with Silybin did not modulate COL1A1 mRNA expression in SCC, a transient reduction was observed for LX2 after 96 h of treatment (*p* < 0.01 with Silybin 7.5 µM). 

Interestingly, we observed a progressive increase in extracellular collagen deposition in SCC treated with FFA that reached the highest production after 144 h (*p* < 0.01), while cotreatment with Silybin counteracted such increase by reducing the extracellular collagen deposition by 40% with 5 µM Silybin (*p* < 0.001 at 144 h) and 30% with 7.5 µM treatment (*p* < 0.05 at 96 h) (Figure 2E). Conversely, the great production of collagen by LX2 monoculture after 144 h of treatment (400% compared to control, *p* < 0.01) persisted even after Silybin cotreatment. Collagen production in the non-NASH environment was not altered by Silybin exposure in SCC, while a significant reduction of 30% with 7.5 µM Silybin after 144 h (*p* < 0.01) was observed in LX2 monoculture. 

### 2.3. Silybin Regulation of Metalloproteinases Activity

Collagen production and deposition in the extracellular compartment is finely regulated at different levels. In particular, the turnover in the extracellular environment is modulated by the protease activity of metalloproteinase 2 and 9 (MMP2-9) [24]. Here we assessed the enzymatic activity of free/active MMP2-9 in the supernatant of SCC, HSCs, and hepatocytes exposed to FFAs and FFA + Silybin and normalized to total proteins present in the cell lysates. FFA treatment induced a ~30% reduction in MMP2-9 activity in Huh7 after 144 h of treatment (*p* < 0.05 vs. CTRL), while treatment with Silybin 7.5 µM restored the activity that was comparable to the control (Figure 3A). A reduction in the MMP2-9 activity by FFA was also observed in the SCC at 96 h (*p* < 0.01 vs. control), which was progressively restored by the presence of increasing concentrations of Silybin (*p* < 0.05 vs. FFA) (Figure 3C). After 144 h, and despite the clear modulation in collagen deposition (Figure 2D); MMP2-9 activity was not modified by the presence of FFA alone. Moreover, a significant reduction of MMP2-9 activity was observed in Silybin treated samples compared to both control and FFA (*p* < 0.01 vs. CTRL; *p* < 0.05 vs. FFA), suggesting that the observed effects on collagen deposition might be the result of some earlier events on MMP2-9 modulation (Figure 3C). Conversely, FFA alone or in combination with Silybin did not exert any effect on MMP2-9 activity in LX2 monoculture (Figure 3B).

### 2.4. Antioxidant Properties of Silybin

The antioxidant properties of Silybin alone or in combination with FFA were evaluated on monoculture and SCC after one hour of treatment. Silybin alone had no effects on ROS production in HSCs or hepatocytes in the mono or cocultures (Figure 4A), while treatment with FFA induced a significant increase of 50% in ROS production independently of the cell culture systems under study (*p* < 0.05 for Huh7 and SCC; *p* < 0.01 for LX2). Cotreatment with 5µM Silybin was particularly efficient in reducing ROS in all the systems with values comparable to those observed in the controls (*p* < 0.05 for Huh7 and SCC; *p* < 0.01 for LX2). Silybin at the highest concentration had a differential effect depending on the cell type, whereas 7.5 µM treatment had no effect on Huh7, a reduction in ROS production was observed in both LX2 and SCC with the highest effect on the coculture (*p* < 0.05 vs. FFA).

## 3. Discussion

Silybin (also referred to as Silibinin) is the major component (50–60%) of Silymarin and is considered the primary bioactive principle that exerts many beneficial effects on the liver. In particular, it has been demonstrated that Silybin has anti-inflammatory, antifibrotic, and antioxidant properties. However, despite its wide use and the extensive studies both in vitro and in vivo, the molecular mechanisms underlying its potential effects as an antifibrotic agent have not been fully elucidated so far.

In this study, we assessed the effect of Silybin in counteracting the FFA noxious stimuli using the SCC approach, which represents a more realistic model for the study of fibrogenesis [23] where hepatocytes and HSC are cultured together in tight contact. The response of each cell type was also analyzed in their respective monocultures. These data confirm our previous results where we reported a differential behavior to the same stimuli when hepatocytes and HSC are in close contact or in the monoculture system [23], highlighting the importance of cell-to-cell contact for the study of the mechanisms involved in fibrosis resolution/reversion. For instance, in the present study, we observed that the cytotoxicity of Silybin alone induced a slight, but statistically relevant, reduction in cell viability of both Huh7 and HSC at the highest concentration (7.5 μM) and longest exposure time (144 h). Conversely, this toxic effect disappeared in the SCC. Even more interesting is the cellular response to FFA, while in Huh7 cell viability was unchanged, in LX2 cells FFA induced a time-dependent proliferative effect, which was slightly reduced by the addition of Silybin. Once again, the SCC treatment with FFA with or without Silybin seemed to be safe in all the experimental conditions.

Regarding the antifibrotic effects, we observed that Silybin had some direct effects on LX2 by reducing cell proliferation as well as a transient regulation of the expression of collagen (COL1A1) after 96 h that could explain the reduction of FFA-induced extracellular collagen deposition observed in the SCC. In a previous in vitro study conducted on primary HSC that were isolated from human liver and activated by just the contact with the flask’s plastic surface, the authors demonstrated that Silybin was able to inhibit PDGF-BB profibrogenic action including cell proliferation, motility, and de novo synthesis of extracellular matrix components [25]. Similarly, in a study conducted on LX2 cells activated by TGF-β, Silybin A demonstrated a significant antiproliferative effect and a strong antifibrotic effect mediated by the downregulation of COL1A1, TIMP1, and MMP2 genes and reduction of TGF-β release [26]. Antiproliferative effects of Silybin on LX2 cells were confirmed by Ezhilarasan et al. that showed that Silybin can modulate pathways involved in cell cycle through upregulation of p27 and p53 and the consequent inhibition of downstream Akt and phosphorylated-Akt protein signaling and Ki-67 protein expression [27]. Conversely, and in spite of the reduction of LX2 cell proliferation and the downregulation of the gene COL1A1 after 96 h, we did not evidence a reduction in the collagen deposition in LX2 monoculture. A possible explanation of this observation is that compared to the data reported in the literature, here we used lower Silybin concentrations. It worth to be mentioned that the reported effects on HSC proliferation were observed using concentrations similar to ours (5–10 µM) silybin [25,27] while the modulation of collagen biosynthesis was assessed only at a higher concentration ranging from 25–100 µM [25,26]. 

To elucidate the molecular mechanisms responsible for the reduction of observed collagen deposition in SCC, we evaluated the activity of MMP2-9 as the principal proteinase involved in collagen degradation. MMPs and their inhibitors—tissue inhibitor of metalloproteinase (TIMPs)—have been extensively studied for their crucial role in fibrogenesis [28,29]. During the fibrogenic process, the great increase of MMP2-9 level is accompanied by the increase in TIMP1-2 level as a mechanism to regulate MMP2-9 proteolytic activity. In this phase, the balance between the two proteins results in MMP2-9 inhibition and a consequent reduction in ECM degradation. On the other hand, during fibrolysis, the reduction of TIMP1, allows free MMPs to degrade ECM [7]. In this study, we evaluated the activity of free-MMP2-9 in the supernatant of the cell cultures and we found that FFAs modulate the proteolytic activity of these enzymes in hepatocytes and that the addition of Silybin at the highest concentration completely restored the enzymatic activity. Conversely, no effects were observed in LX2. On the other hand, in the SCC we found that FFAs affected MMP2-9 activity only at 96 h and that cotreatment with Silybin was effective in restoring the basal activity. Surprisingly, the opposite trend was observed for Silybin cotreatment at 144 h. In the absence of data about the expression of other factors that influence metalloproteinase activity as cytokines expression, any hypothesis about the mechanism that regulates MMP2-9 activity in the SCC system would be fully speculative. 

NAFLD is a multifactorial disease and, among the wide spectrum of molecular events, alterations in the cellular oxidative status have been reported as one of the earliest alterations occurring during the abnormal fat accumulation. Indeed, intracellular lipid storage can induce toxicity enhancing the generation of intracellular reactive oxygen species (ROS), accompanied by an inflammatory response promoting the wound healing process (fibrosis). The antioxidant properties of Silybin have been widely studied, and represent an interesting aspect for the treatment of NAFLD (extensively reviewed elsewhere [30]). We reported previously in an in vivo model of NAFLD that oral supplementation with Silymarin reduced the hepatic Malondialdehyde (MDA) (index of lipid peroxidation) and restored the GSH/GSSG ratio. Considering that the major component of Silymarin is the Silybin, it is likely that the observed effects could be attributed to the latter; however, the contribution of the other components cannot be excluded. In the present study, we explored the antioxidant properties of the Silybin supplementation in the SCC and compared the effect on the monoculture of each cell type to identify which was the target cell. FFA induced enhancement in the production of intracellular ROS both in the monocultures and SCC. The cotreatment with Silybin reestablished the basal redox state in all cell culture systems. Thus, our current findings lead to conclude that the Silybin exerts a general antioxidant effect, which was more pronounced in SCC suggesting a synergy between the two cell types. It is tempting to speculate that oxidative stress is a common feature both in hepatocytes and HSC during the onset of NAFLD pathogenesis. 

The SCC provides useful information about the behavior of the system upon stimuli when hepatocytes and HSC are in close contact. As reported previously [31], the activation of HSC requires cell-to-cell contact with hepatocytes; this effect was induced neither by the soluble mediators released in the transwell system nor by the exposure to hepatocytes conditioned medium. However, as all the in vitro systems, SCC presents some limitations regarding identifying the singular cellular response. In an attempt to tackle this issue, we used the monoculture of each cell type as a control. Moreover, to exclude the effects due to just the interaction of the two cell types in the SCC, we normalized the effects of FFAs or FFA + Silybin vs. the vehicle-treated SCC used as a control. Due to the difference in the response in SCC and monocultures, it is tempting to speculate that the response of SCC is just a summation of the effects from the monocultures. However, the overall response observed in the SCC not always represents the sum (or the average) of each monoculture effect. Taken into consideration all the before mentioned limits, we consider that, under controlled culture conditions, SCC represents a valid in vitro system providing and additional information to the commonly used monoculture set-ups. 

In summary, in the present study, we compared the cellular response when HSC and hepatocytes were cultured separately and together in the SCC. Based on our current data, we hypothesize that FFAs induce a fast redox unbalance in both HSC and hepatocytes, followed by a proliferation of HSC with an increase in collagen synthesis and deposition; likewise, in hepatocytes, FFA induced also a reduction in the activity of MMP2-9 that contributed to the enhanced extracellular collagen deposition observed in the SCC. These effects were restored by the supplementation of Silybin, supporting the potentiality of this compound also as an antifibrotic agent. 

## 4. Materials and Methods 

### 4.1. Chemicals

Cell culture medium: Dulbecco’s modified Eagle’s high glucose medium (DMEM-HG), l-glutamine, penicillin/streptomycin, and fetal bovine serum were purchased from Euro-clone (Milan, Italy). 3-(4,5-dimethylthiazol-2-yl)-2,5-diphenyltetrazolium bromide (MTT), dimethyl sulfoxide (DMSO), oleic acid (C18:1), palmitic acid (C16:0), and phosphate-buffered saline (PBS) were obtained from Sigma Chemical (St. Louis, MO, USA). AlphaLISA cell lysis buffer and Alpha Immunoassay buffer were from Perkin Elmer (Boston, MA, USA). Silybin was obtained from Extrasynthese, Genay, France (Cat#1040). 

### 4.2. Cellular In Vitro Model and Treatment

Hepatic stellate cells (HSCs) LX2 were kindly provided by S.L. Friedman (Mount Sinai School of Medicine, New York, NY, USA). The hepatic cell line Huh7 (JHSRRB, Cat#JCRB0403) were obtained from the Japanese Health Science Research Resource Bank (Osaka, Japan). LX2 and Huh7 were maintained in DMEM-HG supplemented with 100 U/mL penicillin/streptomycin, 2 mM l-glutamine, and 1% or 10% *v*/*v* fetal bovine serum, respectively, at 37 °C in 5% CO_2_ air humidified atmosphere.

The simultaneous coculture (SCC) was freshly prepared in cell ratio 5:1 (hepatocytes:HSCs) for each experiment and maintained in DMEM-HG supplemented with 100 U/mL penicillin/streptomycin, 2 mM l-glutamine, and 1% *v*/*v* fetal bovine serum.

For the induction of NASH, SCC and monoculture were exposed to 1200 µM of free fatty acids (FFA) (oleic:palmitic ratio 2:1 µmol/µmol) [23]. As we reported previously, the minimal Silybin concentration able to induce beneficial effects on FFA-loaded hepatocytes was 5 µM [14], thus we cotreated SCC and monoculture with Silybin 5 or 7.5 µM for 24, 96, and 144 h. Medium containing FFAs and Silybin was refreshed every 2 days until they reach the experimental time point. Effect of Silybin was evaluated also in absence of FFAs to assess possible side effects. For each culture setup (monoculture or SCC), cells exposed to the vehicle were used as a control. The maximal concentration of vehicle (DMSO) was 0.22% *v*/*v*. Cell density was 20,000 cells/cm^2^, 10,000 cells/cm^2^, and 5000 cells/cm^2^ at 24, 96, and 144 h, respectively. Summary of the experimental setup, conditions, and experimental checkpoints are described in Table 1 and Figure 5, respectively. 

### 4.3. Cell Viability

Cytotoxic effect of FFA, Silybin, and the combination of both was assessed by MTT colorimetric assay at 24, 96, and 144 h. LX2 were plated at 10,000, 5000, and 2500 cells/cm^2^; Huh7 20,000, 15,000, and 10,000 cells/cm^2^; and SCC at 15,000, 7500, and 3750 cell/cm^2^ (to test 24, 96, and 144 h, respectively) in a 48-well culture plate and treated as described before. At the experimental time points, cells were incubated for 1 h with medium containing MTT at the concentration of 0.5 mg/mL. Afterward, the medium was removed and formazan crystals were dissolved in 200 µL of DMSO. One-hundred microliters from each well were moved in a microtiter plate, and optical density (OD) was determined at a wavelength of 562 nm on an Enspire^®^ Multimode Plate Reader (Perkin Elmer).

### 4.4. RNA Extraction, cDNA Synthesis, and Gene Expression Analysis by q-PCR

Total RNA was isolated using EuroGOLD RNA Pure according to the manufacturer’s instructions. The total RNA concentration was quantified spectrophotometrically at 260 nm in a Beckman Coulter DU^®^730 spectrophotometer (Fullerton, CA, USA) and purity was evaluated by measuring the ratio A260/A280. Total RNA (1 μg) was reverse transcribed using the High Capacity cDNA Reverse Transcription Kits (Applied Biosystems, Waltham, MA, USA) according to the manufacturer’s suggestions in a Thermal Cycler (Gene Amp PCR System 2400, PerkinElmer, Boston, MA, USA). Quantitative PCR was performed in a CFX connectTM system (Bio-Rad, Hercules, CA USA). All primer pairs were designed using the software Beacon Designer 8.12 (PREMIER Biosoft International, Palo Alto, CA, USA) and were synthesized by Sigma Genosys Ltd. (London Road, UK). Primer sequences are specified in Table 2. 18S and HPRT were used as reference genes. PCR amplification was performed in 15 μL reaction volume containing 25 ng of cDNA, 1 × iQ SYBR Green Supermix, and 250 nM gene specific sense and antisense primers and 100 nM primers for 18S. The data were analyzed using a Bio-Rad CFX manager (version 3.1).

### 4.5. Collagen Quantification 

After 96 and 144 h of treatment, cells were lysed in AlphaLISA lysis buffer (Perkin Elmer) and stored at −80 °C until use. The cell lysate was diluted 1:10 *v*/*v* in Alpha Immunoassay buffer (Perkin Elmer) and collagen content was quantified using COL1A1 AlphaLISA Detection kit (Cat#AL371C, Perkin Elmer) according to the manufacturer’s instructions. Results were obtained fitting the data to a doseresponse sigmoidal curve generated with Graph Pad Prism^®^ version 5.01 (GraphPad Software, Inc., La Jolla, CA, USA) and normalized to the total protein concentration.

### 4.6. MMP2/9 Activity

Metalloproteinase 2 and 9 activity was tested at 96 and 144 h in the supernatant of treated SCC, Huh7, and LX2 with InnoZymeTM Gelatinase activity assay kit (Cat#CBA003, Calbiochem, San Diego, CA, USA). Briefly, 30 µL of supernatant was added to a 96-well plate with 60 µL of activation buffer and 10 µL of substrate working solution. After 2 h of incubation at 37 °C, the developed fluorescence was read at 405 nm (excitation wavelength of 320 nm). Obtained data were fitted to a linear standard curve and normalized to the total protein concentration of the respective cell lysate. 

### 4.7. Total Protein Quantification

Total proteins were quantified by fluorescamine, a nonfluorescent molecule that reacts readily with primary amines in amino acids and peptides to form stable, highly fluorescent compounds. Ten microliters of fluorescamine at 4 mg/mL in DMSO was mixed with 40 µL of cell lysate diluted 1:2 *v*/*v*. Fluorescence was read at 460 nm (excitation wavelength of 390 nm) in an EnSpire^®^ Multimode Plate Reader (Perkin Elmer, Waltham, MA USA). Total protein was calculated using a BSA standard curve dilution.

### 4.8. Quantification of Intracellular Reactive Oxygen Species (ROS)

LX2, Huh7, and SCC were plated in a 96-well plate at the before mentioned cell densities. Cells were exposed for 1 h to FFA and silybin; subsequently, cells were washed with PBS and incubated for 30 min with FBS free culture medium supplemented with 25 mM Hepes/ 20 µM H2DCFDA at 37 °C. Afterward, cells were lysed with fluorescence was quantified by scanning the signal of the attached cells in each well with an Enspire^®^ Multimode Plate Reader (Perkin Elmer) at excitation wavelength 505 nm and emission 525 nm. Fluorescence was normalized by the total proteins present in the cell lysates (µg) assessed using fluorescamine for each sample. 

### 4.9. Statistical Analysis 

Statistical analyses were performed using InStat software Version 3.05 (GraphPad Software, Inc., La Jolla, CA, USA). The Student’s *t*-test was performed for statistical comparison between groups. Value of *p* < 0.05 was regarded as statistically significant.

GraphPad Prism Version 5.0 (GraphPad Software, Inc., La Jolla, CA, USA) software was used to generate graphs.

## Figures and Tables

**Figure 1 molecules-24-01280-f001:**
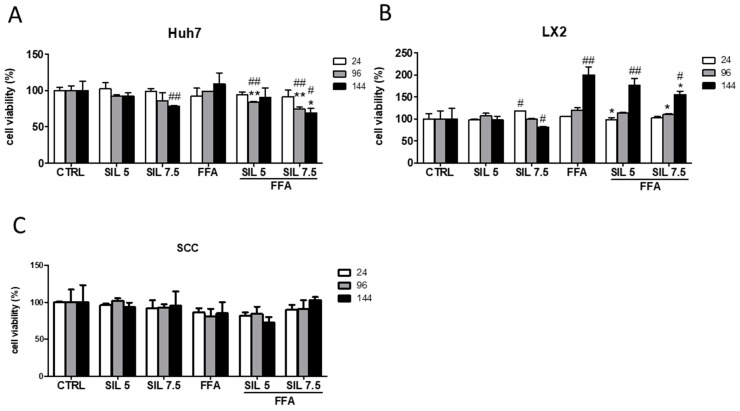
Cell viability and proliferation after Silybin exposure. Cytotoxic effects of Silybin were evaluated in the presence or absence of FFA in Huh7 (**A**), LX2 (**B**), and SCC (**C**) after 24, 96, and 144 h of treatment. # *p* < 0.05, ## *p* < 0.01 vs. CTRL; * *p* < 0.05, ** *p* < 0.01 vs. FFA.

**Figure 2 molecules-24-01280-f002:**
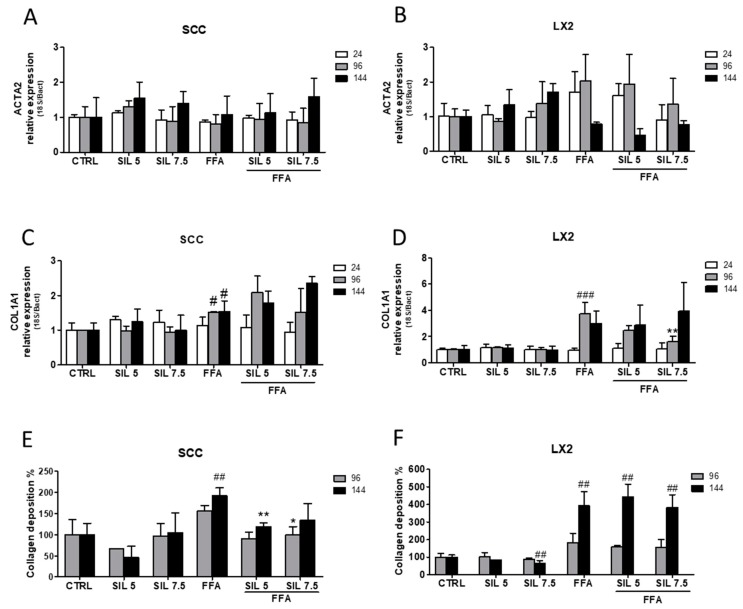
Effect of Silybin on HSC (LX2) activation and collagen production. ACTA2 (α-SMA) gene expression upon FFA and Silybin exposure in SCC (**A**) and LX2 (**B**) vs. control. COL1A1 gene expression upon FFA and Silybin exposure in SCC (**C**) and LX2 (**D**) vs. control. Quantification of the total extracellular collagen upon FFA and Silybin exposure vs. control in SCC (**E**) and LX2 (**F**). # *p* < 0.05, ## *p* < 0.01, ### *p* < 0.001 vs. CTRL; * *p* < 0.05, ** *p* < 0.01 vs. FFA.

**Figure 3 molecules-24-01280-f003:**
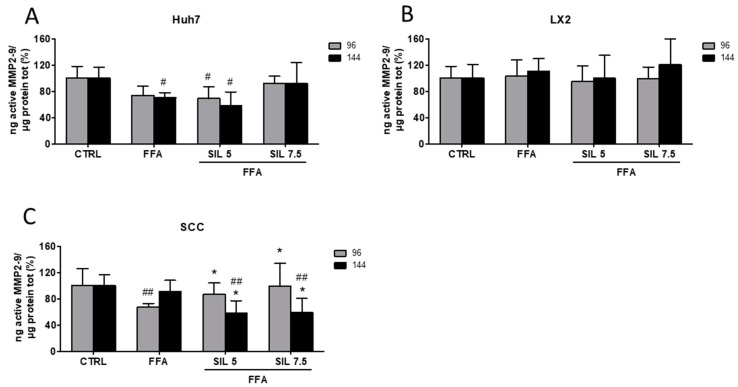
MMP2-9 activity. Enzymatic activity of active MMP2-9 was analyzed in the supernatant of Huh7 (**A**), LX2 (**B**), and SCC (**C**) treated with free fatty acids (FFAs) and Silybin for 96 and 144 h. Activity was normalized to the total proteins in the cell lysate and reported as percentage vs. CTRL sample. # *p* < 0.05, ## *p* < 0.01 vs. CTRL; * *p* < 0.05 vs. FFA.

**Figure 4 molecules-24-01280-f004:**
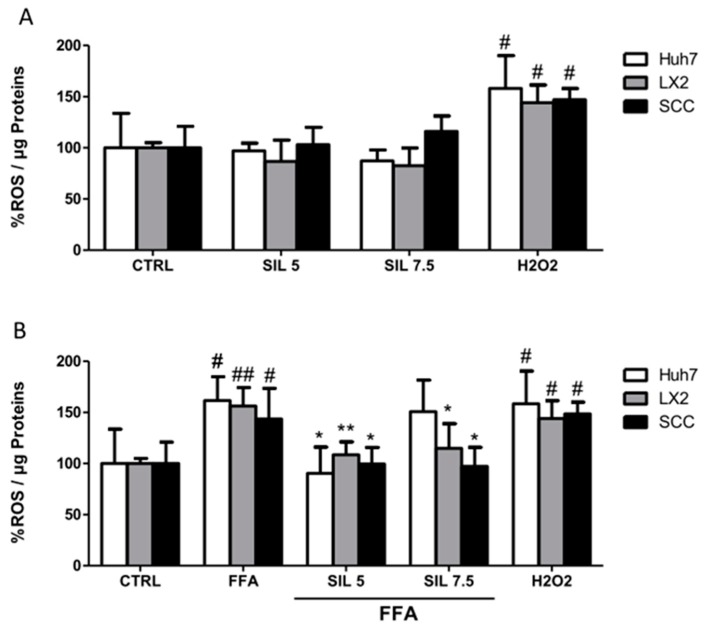
ROS generation. Intracellular ROS was quantified for each condition after 1 h of exposure with Silybin alone (**A**) or in combination with FFA (**B**). Activity was normalized to the total proteins in the cell lysate and reported as a percentage vs. CTRL sample. H2O2 treatment was used as a positive CTRL. # *p* < 0.05, ## *p* < 0.01 vs. CTRL; * *p* < 0.05, ** *p* < 0.01 vs. FFA.

**Figure 5 molecules-24-01280-f005:**
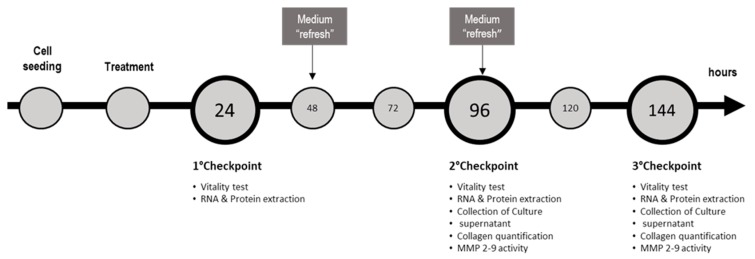
Scheme of the culture proceedings and the experimental checkpoints with the relative determinations.

**Table 1 molecules-24-01280-t001:** Summary of the experimental conditions.

Condition	LX2 Monoculture	Huh7 Monoculture	SCC (Huh7:LX2 – 5:1)
Control	vehicle	vehicle	vehicle
FFA	1200 µM	1200 µM	1200 µM
Silybin	5–7.5 µM	5–7.5 µM	5–7.5 µM
FFA + Silybin	1200 µM + (5–7.5) µM	1200 µM + (5–7.5) µM	1200 µM + (5–7.5) µM

**Table 2 molecules-24-01280-t002:** Primer sequences.

Gene Name	Accession Number	Forward	Reverse
ACTA2 (α-SMA)	NM_001141945	TGTGAATGTCCTGTGGAATTATGC	ACACATAGGTAACGAGTCAGAGC
COL1A1	NM_000088	CGGAGGAGAGTCAGGAAG	ACACAAGGAACAGAACAGTC
18S	NR_003286.2	TAACCCGTTGAACCCCATT	CCATCCAATCGGTAGTAGCG
HPRT	NM_000194	ACATCTGGAGTCCTATTGACATCG	CCGCCCAAAGGGAACTGATAG

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
