# Peer review of "Silybin Modulates Collagen Turnover in an In Vitro Model of NASH"

_molecules, 2019, doi:10.3390/molecules24071280_

Round 1

Reviewer 1 Report

This manuscript set out to study the regulation of Silybin on collagen turnover in an in vitro model of non-alcoholic steatohepatitis (NASH).  The authors used cultured LX2, Huh7 and co-cultured cells to compare the response of FFA and Silybin.  Cell proliferation, collagen formation, MMP2/9 activity, and ROS production were measured. The authors have shown that Silybin inhibited FFA-induced collagen formation, MMP2/9 activity, and ROS production in the co-cultured cells.  The authors concluded that Silybin plays an important role in regulation of the development of NASH through LX2 proliferation, ROS generation and MMP2/9-dependent collagen formation. 

Major points: 

a. In vivo data are missing to demonstrate the effects of Silybin, which greatly dampened the enthusiasm of this manuscript. 

b. The same group have published a similar study with oral application of Silymarin in a rat model of NASH. Silybin is the major active component of Silymarin.  To claim that Silybin reduces NASH through ROS, dose anti-oxidants block the effects of Silybin in vivo? 

Minor points: 

a. typo:

Discussion section, line 1, 'liver'. 

Author Response

Thank you for your comments, please find below our point-by-point answers.

Major points: 

- In vivo data are missing to demonstrate the effects of Silybin, which greatly dampened the enthusiasm of this manuscript.

We did not perform in-vivo studies mainly because several data have been published  supporting the in-vivo anti-fibrotic effects of Silymarin (PMID: 24692823; PMID: 11592601; PMID: 16169303). Despite several evidence on the hepatoprotective effects of Silybin both in pre-clinical models and in clinical settings, very few data are available on the molecular mechanisms underlying the antifibrotic efficacy. Thus, this study aimed to assess in vitro the effect of Silybin in a co-culture model of NASHin the attempt to individuate the effects in the cross-talk between hepatocytes and hepatic stellate cells during fibrogenesis.

b. The same group have published a similar study with oral application of Silymarin in a rat model of NASH. Silybin is the major active component of Silymarin.  To claim that Silybin reduces NASH through ROS, dose anti-oxidants block the effects of Silybin in vivo? 

We previously reported the effect of the oral administration of Silymarin in a murine model of NAFLD. In that study we showed that Silymarin was able to reduce the HFD-induced oxidative stress, by 1) reducing the intrahepatic amount of Malondialdehyde (MDA) which is the most often used index of lipid peroxidation and 2) by restoring the intrahepatic GSH/GSSG ratio. Silybin (also referred as silibinin) is the major component (50-60%) of silymarin and is considered the primary bioactive principle. Therefore it is likely that the reported effects can be attributed to Silybin, however, the contribution of the other components cannot be excluded. The reported effects of Silymarin were obtained from liver homogenates, and thus the data represents the global hepatic redox state. However, it is impossible to study the effects on each cell type. In the present study, we explored the differential effect of Silybin in a monoculture of hepatic stellate cells and when these cells were in contact with hepatocytes.

Minor points: 

a. typo:

Discussion section, line 1, 'liver'. 

Thank you, the typo has been corrected

Reviewer 2 Report

In this manuscript, the authors describe a series of experiments investigating the effect of Silybin in a well-established in vitro co-culture model of early-stage NASH.

As cellular models they used  hepatic stellate cells (HSCs) LX2, and hepatic cells Huh7. Both cell lines were treated as mono- or co-culture (SCC) using a hepatocytes:HSCs ratio of 5:1.

Both mono-cultures and co-cultures were exposed to free fatty acids (FFA) in the absence or in the presence of Silybin. Cell viability, LX2 activation, collagen deposition, metalloproteinase 2 and 9 (MMP2-9) activity and ROS generation were determined at different times (24, 96, and 144 hours).

They found that: (i) exposure to FFA activated LX2 cells (in terms of increase in cell viability and up-regulation of collagen synthesis) and reduced the MMP2-9 activity in Huh7 cells; (ii) co-treatment with Silybin did not affect collagen production in LX2, whereas collagen production was reduced in SCC; (iii) co-treatment with Silybin increased MMP2-9 activity in Huh7 both in mono-culture and in co-culture (SCC); (iv) silybin played antioxidant effects by reducing the FFA-induced production of ROS in all the cell systems.

The authors concluded that silybin exerts its beneficial effects on the liver by reducing activation and ROS generation of HSCs and by increasing MMP2-9 activity in hepatocytes. They suggested that these combined actions of silybin may represent a therapeutic approach to reduce collagen deposition in NASH (antifibrotic activity).

The work is rather original and the scientific question posed by the authors is clear and well defined.

The experimental approach is in general appropriate and described with sufficient details by the authors. The major problem I found is that the use of co-cultures does not allow to understand if the differences in the effects of silybin and fatty acids between the mono-culture and the co-culture depend on the interaction between LX2 and Hu7 cells on if they are only a summation of effects of the single mono-cultures. My suggestion is to repeat experiments using conditioned medium as a control.

The text shows spelling and syntax mistakes.

The quality of figures and tables adequate

On the light of all these criticisms, in the current version the manuscript is not acceptable for publication and requires revision.

Author Response

The work is rather original and the scientific question posed by the authors is clear and well defined.

The experimental approach is in general appropriate and described with sufficient details by the authors. The major problem I found is that the use of co-cultures does not allow to understand if the differences in the effects of silybin and fatty acids between the mono-culture and the co-culture depend on the interaction between LX2 and Hu7 cells on if they are only a summation of effects of the single mono-cultures. My suggestion is to repeat experiments using conditioned medium as a control.

Thank you for your comments. Undoubtedly the in-vitro systems have several limitations and we are fully aware of that. Herein, to exclude the eventual effects due to just the interaction between the two cell types we used co-cultures exposed only to the vehicle as control. With this approach we tried to normalize the effect of the stimulus (FFA or FFA+Sil) vs. the effect induced just by the simple cell-to-cell contact. Regarding a summation of effects from the single mono-cultures, it might be a possibility; however, here we report that cells behave differently when they are in monoculture and when are in co-culture (SCC), and not always the overall response in the SCC represents the sum (or the average) of each mono-culture effect.

The use of hepatocytes conditioned medium (CM) after 24h of FFA exposure on HSC was addressed in the past by our group during the set-up of the SCC model. Due to the short half-life of the soluble pro-fibrotic mediators (for instance active TGF-β half-life is approximately 2 minutes) we did not observe any effect of CM on HSC pro-fibrotic genes (Figure 1, see attached pdf file). This result led us to consider the use of trans-well system where HSC and hepatocytes shared the culture medium to assess the effect of the soluble mediators. Also in this case we did not observe any HSC regulation, whereas the SCC induced a significant HSC activation (α-SMA). This  led us to conclude that for HSC activation cell-to-cell contact was required (data published here PMID: 26187275).

The following paragraph was added in the discussion (line 223):

“The SCC provides useful information about the behavior of the system upon stimuli when hepatocytes and HSC are in close contact. As reported previously [31] the activation of HSC require cell-to-cell contact with hepatocytes, this effect was induced neither by the soluble mediators released in the trans-well system nor by the exposure to hepatocytes conditioned medium. However, as all the in vitro systems, SCC presents some limitations,  as to identify the singular cellular response. In the attempt to tackle this issue, we used the monoculture of each cell type as a control. Moreover, to exclude the effects due to just the interaction of the two cell types in the SCC, we normalized the effects of FFAs or FFA + Silybin vs the vehicle-treated SCC used as a control. Due to the difference in the response in SCC and monocultures, it is tempting to speculate that the response of SCC is just a summation of the effects from the monocultures. However, the overall response observed in the SCC not always represents the sum (or the average) of each monoculture effect. Taken into consideration all the before mentioned limits, we consider that, under controlled culture conditions, SCC represents a valid in vitro system providing and additional information to the commonly used monoculture set-ups”.

The text shows spelling and syntax mistakes.

The text has been deeply edited

The quality of figures and tables adequate

On the light of all these criticisms, in the current version the manuscript is not acceptable for publication and requires revision.

Round 2

Reviewer 1 Report

The authors have replied my previous concerns. Since the effects on ROS production has been previously shown in vivo, although it was treated with Silymarin instead of Silybin, the novelty of this manuscript is greatly dampened.  The priority of this manuscript deserves consideration.